# Role of Nurr1 in Carcinogenesis and Tumor Immunology: A State of the Art Review

**DOI:** 10.3390/cancers12103044

**Published:** 2020-10-19

**Authors:** Peter Kok-Ting Wan, Michelle Kwan-Yee Siu, Thomas Ho-Yin Leung, Xue-Tang Mo, Karen Kar-Loen Chan, Hextan Yuen-Sheung Ngan

**Affiliations:** Department of Obstetrics and Gynaecology, LKS Faculty of Medicine, The University of Hong Kong, Pok Fu Lam, Hong Kong, China; wktpeter@connect.hku.hk (P.K.-T.W.); mkysiu@hku.hk (M.K.-Y.S.); thomashyleung@gmail.com (T.H.-Y.L.); moxil93@connect.hku.hk (X.-T.M.); kklchan@hku.hk (K.K.-L.C.)

**Keywords:** Nurr1, NR4A2, carcinogenesis, cell signaling, immunology

## Abstract

**Simple Summary:**

Nuclear receptor related-1 protein (Nurr1) emerges as a therapeutic target in multiple malignancies and immunotherapies. Previous studies have highlighted its association with clinicopathological parameters, tumorigenesis and therapeutic resistance in cancers. In addition, recent studies unraveled its contribution to the suppression of antitumor immunity, suggesting that inhibition of Nurr1 is a potential method to repress cancer aggressiveness and disrupt tumor immune tolerance. In line with this evidence, the present review provides the roles of Nurr1 in tumor progression and the associated underlying molecular mechanisms. Moreover, the significance of Nurr1 in promoting immune tolerance and potential strategies for Nurr1 inhibition are highlighted.

**Abstract:**

Nuclear receptor related-1 protein (Nurr1), coded by an early response gene, is involved in multiple cellular and physiological functions, including proliferation, survival, and self-renewal. Dysregulation of Nurr1 has been frequently observed in many cancers and is attributed to multiple transcriptional and post-transcriptional mechanisms. Besides, Nurr1 exhibits extensive crosstalk with many oncogenic and tumor suppressor molecules, which contribute to its potential pro-malignant behaviors. Furthermore, Nurr1 is a key player in attenuating antitumor immune responses. It not only potentiates immunosuppressive functions of regulatory T cells but also dampens the activity of cytotoxic T cells. The selective accessibility of chromatin by Nurr1 in T cells is closely associated with cell exhaustion and poor efficacy of cancer immunotherapy. In this review, we summarize the reported findings of Nurr1 in different malignancies, the mechanisms that regulate Nurr1 expression, and the downstream signaling pathways that Nurr1 employs to promote a wide range of malignant phenotypes. We also give an overview of the association between Nurr1 and antitumor immunity and discuss the inhibition of Nurr1 as a potential immunotherapeutic strategy.

## 1. Introduction

Nuclear receptor related-1 protein (Nurr1), also known as NR4A2, belongs to the nuclear receptor (NR) subfamily 4 group A (NR4A). Nurr1 is classified as an ‘orphan’ NR due to the absence of known natural ligands [1]. These NRs are considered as early response genes that respond to many different signals, including fatty acids and hormones [2]. Crosstalk between Nurr1 and various signaling pathways further modulates various cellular and physiological responses, including proliferation, invasion, and apoptosis. Dysregulation of Nurr1 is frequently observed in different types of cancer, with studies demonstrating either pro-oncogenic or tumor-suppressor roles in different contexts [2,3]. Evidence suggesting that suppression of Nurr1 maintains the pluripotency of hematopoietic stem cells supports the contention that the Nurr1 family is important in maintaining terminal differentiation of epithelia [4].

Members of the NR4A family (Nurr1, Nur77, NOR-I) are well-conserved in their genomic organization with ≈91–95% similarity in their DNA binding domain (DBD) and ≈60% similarity in the C-terminal ligand-binding domain (LBD) [2,5]. The N-terminal domain of NR4A members contains an activation function-1 (AF-1), which mediates ligand-independent transactivation of the NR [6]. The DBD is responsible for recognizing the response element and binding to the promoter. Besides, the C-terminal domain is composed of LBD and the ligand-dependent activation function-2 (AF-2) (Figure 1A). The LBD and its amino acid composition define the interaction of the NR with the ligand. Unlike conventional NRs, which have a hydrophobic cleft in the LBD for binding of a coactivator or corepressor, the Nurr1 LBD is tightly packed with hydrophobic side chains, leading to the absence of or atypical ligand-binding cavity clefts [7]. Although the mechanism of regulation of Nurr1-mediated transcription remains unknown, mounting evidence suggests that post-translational and protein-protein interactions are crucial in transcriptional regulation and that the AF-1 domain is involved in ligand recruitment [2,3].

Nurr1 is considered a constitutively active transcription factor due to the constitutive and cell-type-specific activity of its LBD [7]. Nurr1 induces gene expression by binding as a monomer to the nerve growth factor-induced clone B (NGFI-B) response element (NBRE) with common octanucleotide binding site 5′-AAAGGTCA-3′ [2] and as homodimers or heterodimers with Nur77 to the Nur response element (NurRE; 5′-TGACCTTT-n6-AAAGGTCA-3′) [6,8]. Nurr1 also heterodimerizes with retinoid X receptor (RXR) and binds to the direct repeat element with five spacer nucleotide element (DR5; 5′-GGTTCA-n5-AGGTCA-3′) to regulate cell proliferation and survival; this suggests crosstalk of Nurr1 with retinoic acid signaling [9] (Figure 1B). An extensive review of the structure of Nurr1 is summarized by Maxwell et al. [2].

## 2. Expression and Function of Nurr1 in Cancer

Nurr1 promotes or suppresses cancer progression depending on the cellular context, and its oncogenic roles have been reported in many cancers. Overexpression of Nurr1 has been shown to promote cancer cell proliferation, invasion and anchorage-independent growth [10,11]. It also enhances cell survival by suppressing apoptosis [12,13]. Apart from augmenting cancer aggressiveness, Nurr1 confers therapeutic resistance to cancer cells, including radioresistance and chemoresistance to 5-fluorouracil [14,15,16]. Several pro-inflammatory molecules including prostaglandin E_2_ and thromboxane A2 were reported to induce Nurr1 expression, which can lead to carcinogenesis [17,18]. Furthermore, the expression of Nurr1 is correlated with different clinicopathological parameters in multiple cancers [12,13,14]. Since Nurr1 is generally considered to be a transcription factor, its roles in the cytosol are not well characterized. Immunohistochemical staining of Nurr1 revealed its localization in cytosol, and that the level of cytoplasmic Nurr1 correlates with survival and tumor grade in bladder and cervical cancer, respectively [16,19]. In contrast, tumor-suppressive roles of Nurr1 have been described in gastric cancer [20]. Since functions of Nurr1 are cell type-dependent, the roles of Nurr1 in tumorigenesis are summarized and discussed according to specific cancer types.

### 2.1. Breast Cancer

Early studies on the role of Nurr1 in cancer began in 1997 when Maruyama et al. first showed that the NR4A family is closely related to retinoic acid signaling in the breast cancer cell line MCF-7 [21]. Although not many studies on the role of Nurr1 in cancer have been conducted since then, the association of Nurr1 with cancer is supported by some recent work. In 2013, Llpois et al. showed that Nurr1 has a dichotomous role in breast cancer [22]. Stronger Nurr1 staining was observed in normal breast epithelia compared with breast carcinoma cells, with no correlation with grade or stage, but was positively correlated with relapse-free survival. Moreover, there is an inverse correlation between the expression of Nurr1 and p53 in primary cancer. Paradoxically, silencing of Nurr1 attenuates breast tumor xenograft growth and metastasis in vivo. Taken together with its high expression in normal epithelia, Nurr1 may play a role in maintaining a differentiated epithelial phenotype in normal, non-proliferating breast epithelium, yet acquire tumorigenic capability in transformed tissues [22]. Table 1 gives a brief overview of Nurr1 expression and its associated functions and clinicopathological correlation in different cancers.

### 2.2. Bladder Cancer

Nurr1 functions as a transcription factor and localizes in the nucleus. Studies on the localization of Nurr1 and its roles other than that of a transcription factor are sparse. The problem associated with Nurr1 cytoplasmic mislocalization has been addressed by Inamoto et al. [19]. Total Nurr1 expression is shown to positively correlate with stage, grade, and metastasis of bladder cancer; Nurr1 staining in the cytosol is widely observed in primary bladder cancer and barely detected in the normal epithelium. Since Nurr1 is presumed to reside in the nucleus, expression of cytoplasmic Nurr1 is found to correlate with decreased disease-specific survival and recurrence-free survival in patients. Such correlation is not observed when total Nurr1 or nuclear Nurr1 staining is used for analysis. Furthermore, silencing of Nurr1 attenuates the migration of cancer cells. Although the reason for the oncogenic phenotype of Nurr1 in the cytoplasm remains poorly understood, Nurr1 may use mechanisms similar to its homolog Nur77 to exert opposing biological activities. This was demonstrated by Lin et al., who found that Nur77 shuttles from the nucleus to the cytoplasm to interact with Bcl-2 and subsequently switches from an anti-apoptotic to a pro-apoptotic function [27]. Hence, it was suggested that Nurr1 ceases to act as a transcription factor when it translocates to the cytoplasm.

Nurr1’s role as a potential therapeutic target for bladder cancer was the subject of another study by Inamoto et al., which showed that diindolylmethane (DIM)-C-pPhCl (a new class of methylene-substituted DIM) activates the LBD of Nurr1, thereby attenuating the tumorigenic properties of bladder cancer cells [23]. DIM-C-pPhCl induces DNA fragmentation and causes regression of orthotopic bladder cancer in a tumor xenograft model.

### 2.3. Gastrointestinal Cancer

Prostaglandin E_2_ (PGE2) is a pro-inflammatory bioactive lipid produced by colorectal cancer cells that promotes tumor growth by binding to G protein-coupled receptors [17,28]. Vijaykumar et al. demonstrated that PGE2 induces Nurr1 expression and subsequently blocks apoptosis in colorectal cancer cells [17]. Nurr1 was found to be highly expressed in Apc^+/−^ mouse adenomas and sporadic colorectal carcinoma as compared to normal intestinal mucosa. In addition, cyclooxygenase-2 (COX-2), from which PEG2 is derived, is associated with poor prognosis in colorectal cancer and is positively correlated with Nurr1 expression [17,29]. Nurr1 shares similar regions of localization with Ki67 and is mainly found in the proliferation crypt of mice intestinal tissue.

The role of Nurr1 and COX-2 was further studied by Zagani et al. Treatment with a COX-2 inhibitor, parecoxib, downregulates Nurr1 and osteopontin (OPN), a colon cancer progression marker. OPN expression is associated with poor prognosis, lymphatic metastasis, and higher TNM stage in colorectal cancer [30,31]. Although Nurr1 expression increases in tumor tissues, unlike OPN, its expression does not correlate with tumor stage in colorectal cancer [10].

High expression of vascular endothelial growth factor (VEGF) is crucial to tumor growth and angiogenesis and is frequently reported in gastrointestinal cancers [32]. Zhao et al. showed that VEGF induces Nurr1 promoter activity, and mRNA and protein expression, by facilitating binding of cAMP response element binding protein (CREB) to the Nurr1 promoter in endothelial cells [33].

To summarize, Nurr1 is closely related to COX-2, OPN, and VEGF. Aberrant expression of all or either one of these proteins may dysregulate the expression of their interacting partners and act synergistically to promote inflammation to promote development of gastrointestinal cancers.

Unlike many other gastrointestinal cancers, Nurr1 is reported to be tumor-suppressive in gastric cancer [20,24]. Chang et al. reported that Nurr1 is significantly downregulated in primary gastric cancer compared with the normal gastric mucosa; it is also downregulated in synchronous liver metastasis compared with the paired gastric cancer [24]. Another study by Misund et al. demonstrated that endogenous Nurr1 enhances apoptosis and attenuates gastrin-induced invasion [20]. Nurr1 is shown to be negatively regulated by two gastrin-induced proteins: (i) inducible cAMP early repressor (ICER), which represses Nurr1 transcription; and (ii) zinc finger protein 36, C3H1 type-like 1 (Zfp36l1), which degrades and reduces Nurr1 mRNA levels. Silencing of Nurr1 promotes cancer cell migration and the effect is further augmented in response to gastrin treatment. Similar to what is observed in bladder cancer, as discussed above, Nurr1 cytoplasmic localization is observed in gastric cancer by gastrin-mediated nucleus-cytosol shuttling, but the role of cytoplasmic Nurr1 remains to be elucidated. In contrast, Han et al. demonstrated that ectopic overexpression of Nurr1 enhances gastric cancer formation in vivo [14]. It also confers chemoresistance to 5-fluorouracil by attenuating 5-fluorouracil-induced apoptosis. High Nurr1 expression is associated with unfavorable prognosis, particularly in those receiving chemotherapy. Therefore, the role of Nurr1 in gastric cancer remains controversial and more functional characterization is needed to understand its contribution to gastric carcinogenesis.

### 2.4. Lung Cancer

Thromboxane A2 (TXA2), like PGE2, is another COX-2 derived product. An agonist of TXA2 receptor (TP), called I-BOP, is found to induce expression of Nurr1 significantly, but not that of other NR4A members, Nur77 and NOR-1 [18]. Silencing of Nurr1 attenuates cell proliferation by downregulating I-BOP-induced cyclin D1 expression and the basal thymidine uptake. Apart from TXA2, COX-derived PGE2 (briefly discussed previously) is shown to induce Nurr1 expression in some, but not all lung carcinoma cell lines via cAMP/PKA dependent pathways.

### 2.5. Cervical Cancer

Ke et al. compared the expression profile of cervical cancer cell line HeLa with HeLaHF (a non-transformed revertant of HeLa) and found that all three members of the NR4A family are downregulated in HeLaHF, suggesting that NR4A members are potential oncogenes [11]. Silencing of *Nurr1* was shown to reduce anchorage-independent growth by increasing anoikis, as demonstrated by the increase in DNA fragmentation. Apart from inhibiting apoptosis, Sun et al. reported that Nurr1 increases cell proliferation [25]. Notch activation (by overexpression of four Notch receptors: ICN1, 2, 3, 4) suppresses tumor growth in cervical cancer with ICN1 displaying the most significant antitumor effect. Overexpression of ICN1 reduces Nurr1 and induces p63 expression. Transient transfection of Nurr1 in cervical cancer overexpressing ICN1 abolishes ICN1-induced cell growth arrest and ICN1-induced p63 expression, suggesting that Nurr1 inhibits Notch-mediated tumor suppression.

### 2.6. Prostate Cancer

Nurr1 expression is correlated with the clinicopathological features of prostate cancer, including the Gleason score and TMN classification, though a correlation with age is not observed. Knockdown of *Nurr1* by small interfering RNA (siRNA) induces apoptosis and inhibits cell proliferation and migration [12].

### 2.7. Pancreatic Cancer

Nurr1 is a significant independent prognostic indicator in patients with pancreatic ductal adenocarcinoma. High Nurr1 expression in this cancer has been correlated with histological subtypes with higher grade, higher stage, high Ki-67 expression, and poor overall survival [13]. In addition, functional studies revealed that silencing of Nurr1 promotes apoptosis and attenuates cell migration, invasion, and proliferation.

### 2.8. Brain Cancer

Nurr1 is highly expressed in multiple glioblastoma cell lines and patient-derived cancer cells, as compared to Nur77 and NOR-I [26]. Knockdown of Nurr1 by antisense oligonucleotides reduces cell proliferation, invasion and survival in cell lines and patient-derived cells. Treatment with the Nurr1 antagonist DIM-C-pPhCl abolishes cell aggressiveness, induces cell apoptosis and suppresses tumor formation in nude mice. High expression of Nurr1 is significantly associated with shorter disease-free survival. Interestingly, the natural compound *n*-butylidene phthalide (BP), isolated from the chloroform extract of *Angelica sinensis*, upregulates the expression of Nurr1, Nur77 and NOR-I [34]. The induced Nur77 was reported to promote apoptosis and tumor regression, suggesting that the antitumor activity of BP may be mediated by the NR4A family.

### 2.9. Hematological Cancers

Nur77 and NOR-1 were reported to be critical tumor suppressor proteins in acute myeloid leukemia [35]. Nur77/NOR-1-double knockout mice are associated with myeloid leukemogenesis, abnormal expansion of myeloid progenitor and hematopoietic stem cells, malfunctional extrinsic apoptotic pathway and reduced expression of the AP-1 transcription factor. In addition, reduction in the gene dosage of Nur77 and NOR-1 in hypoallelic (Nur77^+/−^NOR-1^−/−^ or Nur77^−/−^NOR-1^+/−^) mice beyond a critical threshold is also sufficient to cause a mixed myelodysplastic/myeloproliferative disease with progression to acute myeloid leukemia [36]. Moreover, significant downregulation of Nur77 and NOR-1 was observed in chronic lymphocytic B-cell leukemia, follicular lymphoma and diffuse large B-cell lymphoma [37]. Restoration of Nur77 or NOR-1 promotes apoptosis and tumor regression in aggressive lymphoma. In addition, Nurr1, Nur77 and NOR-1 were reported to be highly expressed in prednisolone-resistant acute lymphoblastic leukemia, yet they do not functionally contribute to the resistance [38].

## 3. Signaling Pathways Regulating Nurr1 Expression

Aberrant signaling pathways are observed in the transformation and maintenance of malignant phenotypes of tumors. Since Nurr1 is frequently associated with either oncogenic or tumor-suppressive properties in different contexts, it is logical to propose that dysregulation of signaling pathways affects Nurr1 expression and promotes cancer development. The promoter of Nurr1 contains a highly conserved sequence of cAMP response element (CRE) and kappa B (κB) site. Mounting evidence indicates that multiple signaling events eventually converge to promote the phosphorylation of CRE binding protein (CREB) and nuclear factor kappa B (NF-κB) to enhance the transcriptional activity of Nurr1 [39]. Although transcriptional regulation is considered as the primary method to regulate Nurr1 expression, post-transcriptional regulation by microRNA (miR) also exerts an impact on the translation of Nurr1 mRNA [40].

The interplay of Nurr1 with the TXA2 pathway, PGE2 pathway, VEGF, and miR at the transcriptional and posttranscriptional levels will be summarized in the following section. Figure 2 and Table 2 summarize the interplay of Nurr1 with different signaling pathways in cancers.

### 3.1. TXA2 Pathway

Aberrant TXA2-TP signaling axis has been reported in multiple cancers. TPα and TPβ are the two isoforms of TP and they couple to guanine nucleotide-binding protein (G protein) for functional mediation [42,43]. TPα couples to G_s_ and G_q_, whereas TPβ couples to G_i_ and G_q_ [43]. Overexpression of TXA2, a COX-2-derived product, and activation of TP promotes cell survival and growth of melanoma, prostate and urothelial cancer [18,41,44,45]. 

The TXA2-TP signaling pathway can be elucidated as follows: I-BOP, a TP agonist, is found to induce Nurr1 expression in three ways [18]. (1) cAMP/protein kinase A (PKA)/CREB pathway: Upon TP activation, TPα couples to Gs to generate cAMP, which subsequently activates PKA and CREB; (2) ERK/CREB pathway: I-BOP induces MEK and ERK phosphorylation, possibly via Ras activation, leading to CREB activation; (3) PKC/CREB pathway: When TP is activated, TPβ links to phospholipase C (PLC) via coupling to Gq to activate protein kinase C (PKC), which further phosphorylates CREB. Meanwhile, PKC but not PKA is found to promote TP-mediated ERK1/2 phosphorylation, thereby suggesting that expression of I-BOP-induced Nurr1 may be partly due to activation of ERK by PKC.

### 3.2. PGE2 Pathway

PGE2 has been linked to cell proliferation, metastasis, cancer stemness, and chemoresistance in multiple cancers, including colorectal, endometrial, and lung cancer [46,47,48,49]. It mediates diverse and unique signaling networks via four G-protein-coupled receptors (EP1–EP4) for distinct second messenger pathways [17,39]. Similar to TXA2, PGE2 induces Nurr1 expression via cAMP-dependent and cAMP-independent pathways: (1) cAMP-dependent pathway: Upon activation by PGE2, EP2 couples to G_s_, resulting in sequential cAMP activation, PKA activation, and CREB phosphorylation to initiate transcription of *Nurr1* [18,49]; (2) cAMP-independent pathway: EP1 activated by PGE2 couples to G_12/13_, resulting in Rho activation [50]. Activated Rho then phosphorylates I-κB, leading to phosphorylation and dissociation of PKAc from the I-κB/NF-κB/PKAc complex and subsequent PKA-c-dependent phosphorylation of NF-κB and CREB to induce Nurr1 transcription [39]. In addition, activated EP1 also couples to G_i_ to upregulate hypoxia-inducible factor-1 alpha (HIF1-α) via the PI3K/Akt/mTOR signaling pathway [51].

### 3.3. P53/miR-34/Nurr1 Loop

p53 is one of the most studied tumor suppressor proteins in cancer that stabilizes various genotoxic and cellular stresses, including DNA damage, oncogenic activation, hypoxia, and nutrient deprivation through transcription-dependent and -independent mechanisms [52,53]. Mounting evidence suggests that miRs are important components in the p53-dependent tumor suppressor network. MiR-34 (with isoforms miR34a and miR34b/c) is the direct transcriptional target of p53 and plays an important role in inducing apoptosis and cell cycle arrest by post-transcriptional regulation of miR-34 responsive genes [54,55,56,57]. MiR-34 has been shown to regulate and suppress Nurr1 expression by binding to the 3′UTR of Nurr1 mRNA [40,57,58]. Reduced Nurr1 protein expression mitigates its interaction with p53, thereby forming a tumor-suppressive p53/miR-34/Nurr1 loop. Nonetheless, when there is dysregulation of expression of any member in the loop, for instance, loss of p53 expression leads to decreased miR-34 expression and subsequent upregulation of Nurr1. This creates a positive feedback mechanism that shifts the regulatory loop from being tumor suppressive to oncogenic.

### 3.4. VEGF/Protein Kinase D (PKD) Pathway

Angiogenesis, a hallmark of cancer, is important for supplying nutrients to tumor masses for growth and metastasis [59]. These tumor blood vessels are characterized by immaturity and impaired functionality and offer survival advantages to tumor tissues, such as providing a hypoxic tumor microenvironment and reducing infiltration of immune cells [60]. The efficacy of radiotherapy, which relies on reactive oxygen species generation, and delivery of chemotherapeutic drugs have been hindered by hypoxic conditions and vessel immaturity, respectively [33]. Among various growth factors secreted by the tumor and stromal cells, VEGF plays a critical role in neovascularization. Evidence suggests that VEGF induces Nurr1 promoter activity and expression in endothelial cells. Silencing of Nurr1 suppresses endothelial cell proliferation, migration, and matrigel angiogenesis. VEGF-induced Nurr1 expression is mediated by VEGF receptor 2 (VEGFR2). It phosphorylates PKC, PKD, and CREB in a sequential manner [33,61,62]. Activated CREB then binds to the Nurr1 promoter to initiate transcription. In addition, VEGF also modulates NF-κB activation but with a moderate effect, as deletion of the upstream κB site moderately reduces Nurr1 promoter activity [33]. Taken together, VEGF induces Nurr1 expression primarily via a VEGF2-mediated PKC-dependent PKD axis.

## 4. Crosstalk of Nurr1 with Pro-Tumorigenic and Tumor-Suppressive Molecules

There is an extensive crosstalk of Nurr1 with pro-tumorigenic or tumor-suppressive signaling. Nurr1 has been reported to regulate cancer aggressiveness, self-renewal, survival, DNA repair and metabolism. Downstream signaling of Nurr1 in relation to the diverse phenotypes in cancers will be discussed. The recent identification of Akt and ERK as downstream targets of Nurr1 in carcinogenesis by our group will also be covered. Figure 3 summarizes the interplay of Nurr1 with different signaling pathways in cancers.

### 4.1. PI3K/Akt/mTOR and MEK/ERK Pathways

Our group recently reported that human papillomavirus (HPV)-induced Nurr1 independently promotes activation of PI3K/Akt/mTOR and MEK/ERK signaling cascades [16]. PI3K/mTOR dual inhibitor dactolisib and MEK inhibitor trametinib abolish Nurr1-augmented cell proliferation, self-renewal, and invasion. Akt signaling is also indispensable in mediating Nurr1-conferred radioresistance. Since Akt and ERK primarily reside in the cytosol and there is an accumulation of Nurr1 in the cytosol in primary cervical cancer, it is postulated that Nurr1 translocates from the nucleus to the cytosol to promote Akt and ERK activation, thereby inducing malignant behaviors. Alteration of molecular functions after translocation is supported by cytoplasmic Nur77, which induces conformational changes in Bcl-2 and converts anti-apoptotic Bcl-2 into a pro-apoptotic factor, triggering cytochrome c release and apoptosis [27,63]. In contrast, Smith et al. reported that MEK1/2 inhibitor suppresses transcriptional activity of Nurr1 [64]. The activity is significantly augmented in BRAF-V600E melanoma cells.

### 4.2. Wnt/β-Catenin Signaling

Dysregulation of Wnt signaling is associated with cancer development and stemness [65]. PGE2 stimulation has been reported to promote activation of the Wnt pathway [66]. OPN is one of the transcriptional targets of the aberrant Wnt/β-catenin pathway [67]. Downregulation of OPN by COX-2 inhibitor treatment is mediated by the downregulation of Nurr1 in the early stage and the inhibition of Wnt signaling in the late stage, as determined by the reduced expression of β-catenin and other Wnt/β-catenin targets [10,31]. The association between Nurr1 and OPN was confirmed by the luciferase reporter study, which showed that Nurr1, as a monomer, directly binds to and transactivates the OPN promoter [68]. The transactivation is dependent on the AFs and does not require heterodimerization with RXR.

Mutation and overexpression of β-catenin are frequently observed in activated Wnt signaling pathways in cancers. Its level is regulated by the β-catenin destruction complex, particularly by the adenomatous polyposis coli (APC) protein [69]. Nurr1 represses β-catenin-mediated transactivation and, hence, β-catenin signaling [70]. The repression is mediated by the DBD of Nurr1 without involving AF-1 and AF-2. Overexpression of Nurr1 significantly reduces the expression of Axin2, a downstream target of β-catenin and a negative regulator of the Wnt signaling pathway, leading to activated Wnt signaling. Meanwhile, β-catenin represses the transcriptional activity of Nurr1 in osteosarcoma and cervical cancer with the involvement of the LBD of Nurr1 [70]. Interestingly, the transcriptional activity is enhanced in 293T cells, suggesting that the crosstalk between Nurr1 and β-catenin may be cell type-dependent.

### 4.3. P53

Nurr1 promotes cell survival and possesses anti-apoptotic properties in colon cancer HCT116 p53^+/+^ cells, but not in HCT116 p53^−/−^ cells with or without doxorubicin treatment [58]. It is suggested that Nurr1 protects cells from doxorubicin-induced apoptosis in a p53-dependent manner. Besides, Nurr1 suppresses p53 self-assembly and impairs the transcriptional activity of p53 in an interaction- and dose-dependent manner [58]. This is mediated by the interaction between DBD of Nurr1 and the -COOH terminus of p53 containing the tetramerization domain. Furthermore, expression of Nurr1 is inversely correlated with p53 expression in primary breast cancer [22]. Taken together, Nurr1 negatively affects p53 expression by promoting the Nurr1/p53/miR-34 positive feedback mechanism and suppressing p53 transcriptional activity to promote development of cancer.

### 4.4. DNA-Dependent Protein Kinase (DNA-PK)

DNA damage is considered as one of the primary causes of cancers. DNA-PK is an important component of the DNA repair mechanism, which responds to the DNA damage and governs genome integrity [71]. The DNA-PK is composed of a catalytic subunit, DNA-PKcs, and a heterodimer of Ku proteins (Ku70/Ku80). Nurr1 interacts with DNA-PKcs and translocates to the double-strand break foci [72]. Phosphorylation of Nurr1 by DNA-PKcs is found to be critical for DNA repair, whilst DNA-binding and the transcriptional activity of Nurr1 are not required by the repair machinery. Silencing of Nurr1 or mutation of *DNA-PKcs* inhibits DNA repair after irradiation in transformed cells. Double-strand break repair is further impaired when both Nurr1 and Nur77 are downregulated.

### 4.5. Fatty Acid Oxidation Pathway

The metabolic shift to fatty acid oxidation provides an alternative energy source and promotes cell survival in many malignancies [73]. The association of Nurr1 with fatty acid metabolism was reported by Holla et al. [74]. Nurr1, as a transcription factor, recruits transcriptional coactivator peroxisome proliferator-activated receptor gamma coactivator 1-alpha (PGC-1α) and steroid receptor coactivator 1 (SRC-1). The transcriptional complex then binds to the NBRE sites of the fatty acid oxidation genes, including acyl-CoA oxidase and fatty acid-binding protein 2 (FABP2) and regulates their transcription. Induction of Nurr1 by PGE2 or ectopic expression of Nurr1 increases fatty acid oxidation and this effect is abolished when Nurr1 is silenced, confirming PEG2 and Nurr1 as potential regulators of metabolic reprogramming, which has been observed in different malignancies.

### 4.6. Transforming Growth Factor-Beta (TGF-β) Pathway

Aberrant TGF-β signaling is associated with cancer progression and metastasis [75]. The inflammatory cytokines interleukin-1 beta (IL-1β) and tumor necrosis factor-alpha (TNF-α) were reported to induce Nurr1 expression, which, in the presence of TGF-β, potentiates SMAD activation of cancer development [76]. These responses are also significantly augmented when Nur77 is overexpressed. Nur77 promotes TGF-β/SMAD-induced epithelial-to-mesenchymal transition and invasion by interacting with and promoting AXIN2-RNF12/ARKADIA-induced SMAD7 degradation to enhance the expression of activated TGF-β type I receptor (TβRI) in breast cancer. This suggests Nurr1 and Nur77 are important in responding to inflammatory stimuli by activating TGF-β signaling for cancer progression.

## 5. Nurr1 and Inhibition of Antitumor Immunity

### 5.1. Treg-Mediated Immunosuppression

Regulatory T cells (Tregs) promote tumor development and progression by suppressing antitumor immunosurveillance and providing an immunosuppressive tumor microenvironment. Tregs, characterized by CD4, CD25, and Foxp3 expression, are critical to maintain tolerance to self-antigens and prevent autoimmune diseases [77]. High infiltration of Tregs and a low ratio of cytotoxic CD8^+^ T cells to Tregs have been associated with poor prognosis in multiple cancers [78]. For instance, accumulation of Tregs is associated with gastrointestinal inflammation, tumor progression, and induction of pro-angiogenic marker VEGF in breast and endometrial cancers [32,79].

Growing evidence suggests that there is a close association between Nurr1 and immunosuppression, immune homeostasis, and Treg development. Sekiya et al. reported that the expression level of Nurr1 is higher in Foxp3^+^ Tregs as compared to Foxp3^-^ Tregs and T cell receptor (TCR) stimulation transiently induces Nurr1 expression in naive T cells. Ectopic expression of Nurr1 induces Foxp3 expression by acting primarily on the proximal promoter and moderately on the intronic enhancer, designated as conserved noncoding sequence 1 (CNS1) [80,81]. Besides, Nurr1 affects the epigenetic status of the Foxp3 promoter/enhancer and induces Treg-like epigenetic modification i.e., active histone modifications, histone H4-acetylation, and histone H3K4-trimethylation without inducing demethylation of CNS2, as observed during Treg development. In addition, Nurr1 is able to repress interferon-γ (IFNγ) and Th1 differentiation by both Foxp3-dependent and Foxp3-independent pathways and induce immunosuppression by downregulating interleukin (IL)-2 and upregulating CD25. T cell-specific Nurr1 deletion causes exacerbation of colitis as a result of aberrant Th1 induction and abortive Treg induction; this confirms the importance of Nurr1 in maintaining immune homeostasis. Furthermore, Treg-specific *Nurr1* deletion confirms that Nurr1 plays a critical role in maintaining Treg lineage by sustaining Foxp3 expression via interactions with Runx/CBFβ complexes. Nurr1, by upregulating CD25, transforming growth factor-beta 1 (TGF-β1), and C-C motif chemokine receptor 9 (CCR9), is also indispensable for mediating the maximal suppressive actions of Tregs; this indicates that Nurr1 is a Treg regulator.

In 2018, Hibino et al. showed that NR4A factors are highly expressed in the potent immunosuppressive effector Tregs and such Treg-specific NR4A deletion disrupts Treg-mediated immune tolerance and promotes antitumor immunity [82]. Tumor growth is suppressed in a xenograft Nur77/Nurr1-double knockout mouse model and the expression of Treg-signature genes, including Foxp3 and CTLA-4, is attenuated, without development of autoimmunity, but Nur77- or Nurr1-single knockout mice are not able to delay or inhibit tumor growth [82]. Spontaneous inflammation was observed in Nurr1/Nur77/NOR-1-triple knockout clones, which indicates the importance of systemic NOR-1 in suppressing autoimmunity. In addition, tumor-draining lymph nodes in the Nur77/Nurr1-double knockout mice show downregulation of Foxp3^+^ Tregs and an accumulation of CD8^+^ T cells with higher expression of Ki-67 compared to the wild-type mice [82]. In the tumor tissue, though the number of Foxp3^+^ Tregs does not decrease, CTLA-4 expression on Tregs is downregulated and higher fractions of effector CD8^+^ cytotoxic T cells, IFNγ^+^CD4^+^ T cells, and CD80^+^ dendritic cells are observed, highlighting the notion that NR4A inhibition breaks the immune tolerance against cancer cells [82].

### 5.2. COX-2/Nurr1 Axis in Immunosuppression

As discussed above, COX-2 and PGE2 are constitutively overexpressed in multiple cancers so as to promote and maintain the malignant phenotypes [83]. Besides, they help tumors evade immunosurvillence [84]. PGE2 stimulation is shown to upregulate Nurr1 in Tregs in a dose-dependent manner [82]. As expected, treatment with COX-2 inhibitor downregulates expression of Nurr1 target genes lkzf4 and Foxp3 in Tregs [82]. Apart from COX-2 inhibitor, DNA topoisomerase I inhibitor, camptothecin, has been shown to suppress NR4A transcriptional activity. Under in vitro helper T-cell differentiation conditions, camptothecin suppresses the induction of induced Tregs (iTregs) and promotes the induction of IFNγ^+^ Th1 cells in a TGF-β-independent manner. Although camptothecin does not alter T cell populations, including Tregs, in the periphery and the thymus, it destabilizes Foxp3 expression in Tregs both in vitro and in vivo. Combined therapy with COX-2 inhibitor and camptothecin synergistically deplete Tregs and enhance Ki67^+^CD8^+^ effector T cells in the lymph nodes. The treatment also increases tumor-infiltrating IFNγ-producing CD4^+^ and CD8^+^ T cells and inhibits tumor growth in vivo [82]. Adoptive transfer of wild-type Tregs but not Nurr1/Nur77-double knockout Tregs in tumor-bearing mice diminishes the antitumor effects conferred by the drugs; this further confirms that the pharmacologic effect is dependent on the inhibition of NR4A-mediated Treg functions.

### 5.3. Suppression of Anti-Leukemia Immunity

In a study on the interaction between hematopoietic stem and progenitor cells (HSPCs) and leukaemia mesenchymal niche, mesenchymal stromal cells (MSCs) derived from Fanconi anaemia (FA) patients with acute myeloid leukaemia (AML) (FA-AML MSCs) promoted expansion of healthy donor HSPCs, myeloid expansion of bone marrow CD34^+^ cells, and Treg differentiation. The HSPC/myeloid expansion is mediated by leukemic mesenchymal COX-2-prostaglandin (PG) secretome [85]. Three pathways have been identified that mediate the effect of the COX-2 PG secretome: (i) NR4A2 signaling, (ii) Wnt signaling, and (iii) Treg function pathways.

Downregulation of PG by COX-2 inhibition ameliorates HSPC/myeloid expansions and reduces expression of NR4A and expression of Treg genes (Foxp3 and CTLA-4) in AML MSCs cocultured with CD34^+^ populations, indicating an association between the COX2-PG secretome and NR4A-Treg signaling [85]. In addition, Nurr1 and Nur77 act synergistically on the promoter of proto-oncogene β-catenin (CTNNB1) to enhance Wnt activation [85]. CTNNB1 knockdown significantly upregulates CD8^+^CCR7^+^ T effector cells, an important subset in anti-leukemia immunity, without affecting the Treg population. The CD8^+^CCR7^+^ T cells also exhibit more potent cytotoxic activity against leukemic cells. Taken together, the interaction between the leukemic mesenchymal COX2-PG secretome and HSPCs induces the NR4A/Wnt/Treg signaling axis, which in turn attenuates anti-leukemia immunity.

### 5.4. Treg Development

Nurr1 is crucial for thymic Treg development. Compound deletion of all NR4A genes (Nur77, Nurr1, and NOR1) in T cells impairs Treg development and induces Th2-type inflammation, resulting in lethal systemic autoimmunity [81]. Foxp3^+^ Tregs are also barely detectable in the thymus, spleen, and lymph node in NR4A triple-knockout mice. Nonetheless, Nur77/NOR-1-double knockout mice are able to develop a substantial Treg population in the thymus, indicating the critical contribution of Nurr1 to thymic Treg development. Furthermore, Nurr1 acts as an effector to determine the fate of thymic CD4^+^ T cells by ‘translating’ the strength of TCR signaling to the transcriptional control of Foxp3. It is demonstrated that activation of Nurr1 induces Foxp3 for Treg development and stronger Nurr1 activation induces cell death by negative selection [81].

### 5.5. T Cell Exhaustion

T cell exhaustion, defined as an acquired state of dysfunctional T cells, is commonly observed in cancer or chronic infections [86]. Exhausted CD8^+^ T cells are poorly cytotoxic and express distinctive patterns of inhibitory receptors, including PD-1, TIM-3, and LAG-3 [87,88]. Although some checkpoint blockades restore functions of exhausted CD8^+^ T cells, mechanisms underlying T cell exhaustion remain poorly understood [89,90]. Accumulating evidence indicates that exhausted T cells exhibit a distinctive epigenetic profile with an exhaustion-specific chromatin accessibility pattern [86,91,92]. In 2016, Sen et al. used transposase-accessible chromatin with high throughput sequencing (ATAC-seq) in T cells in response to lymphocytic choriomeningitis virus (LCMV) infection to demonstrate that differentiation of naïve CD8^+^ T cells undergoes large-scale chromatin remodeling associated with a net increase in chromatin accessibility [86].

In 2017, Mognol et al. compared chromatin accessibility of exhausted tumor-reactive OT-I and non-exhausted tumor-ignorant P14 cytotoxic T lymphocytes and identified the NBRE motif as the most enriched consensus binding motif in OT-I specific regions [91]. Gene expression profiling shows that *Nurr1* and *Nor1* are strongly upregulated in OT-I cells. *Nurr1* loci are also more accessible in the exhausted T cells. In addition, enrichment of the consensus binding motif for transcription factors associated with T cell activation i.e., NFAT: AP-1 composite site, and consensus binding motif for NFAT without an AP-1 site are observed. In contrast, the binding motif for NFAT without AP-1 is significantly less enriched in non-exhausted T cells, lending credence to the study by Martinez et al., who found that NFAT induces T cell exhaustion in the absence of AP-1 [93]. Furthermore, a region located −22.4 kb 5′ of the Pdcd1 locus transcription start site (TSS) is selectively accessible in exhausted T cells in both tumor and LCMV models [91,92]. Although NFAT does not bind to this region, an engineered constitutively active version of NFAT-1 with three amino acid mutations, which prevents AP-1 binding (CA-RIT-NFAT1), partially drives PD-1 expression, proving that the region is functionally active [86]. By sequence analysis, three potential NBRE motifs and two NFAT motifs were identified in the regions. Given that NFAT does not bind to this region, NFAT may make the region more accessible to Nurr1, NOR-1, and other transcriptional factors, which perpetuates the exhaustion state [91]. Taken together, Nurr1 acts as a regulator of CD8^+^ T cell exhaustion by acting on the NBRE motif or in cooperation with NFAT. Constitutively active NFAT, under conditions which are not favorable for its interaction with AP-1, participates in the transcriptional program that maintains T cells in a state of exhaustion, by increasing chromatin accessibility to Nurr1, NR4A members, and other transcriptional factors during chronic viral and tumor challenges [91,92].

### 5.6. M2 Macrophage Repolarization

Macrophage polarization enables macrophages to switch between M1- and M2-subtypes. The M1 macrophage is designated as the classically activated macrophage, which is involved in proinflammatory and antitumor responses. M2 macrophage, known as an alternatively activated macrophage, antagonizes inflammatory responses and maintains an immunosuppressive tumor microenvironment. Mahajan et al. showed that Nurr1 enhances expression of the M2 macrophage marker CD36 on bone marrow-derived macrophages (BMDMs) and increases expression of M2 prototype genes, such as arginase 1, which is a direct transcriptional target of Nurr1 [94,95]. Overexpression of Nurr1 in BMDMs also increases secretion of anti-inflammatory cytokines (IL-10) and decreases secretion of tumor necrosis factor-alpha (TNF-α). These data highlight the fact that by repolarizing macrophages to the M2 type, Nurr1 plays an anti-inflammatory role.

### 5.7. Suppression of Immunotherapy

The limited clinical efficacy of chimeric antigen receptor (CAR) T cell therapy against solid tumors may partly be attributed to their acquired exhaustion state upon chronic antigen stimulation. Genomic and epigenetic analysis of highly exhausted PD-1^hi^TIM3^hi^ CAR and endogenous CD8^+^ tumor-infiltrating lymphocytes (TIL) revealed that they exhibit similar gene expression and chromatin accessibility profiles with enrichment of consensus NBRE and NFAT motifs in the accessible regions in PD-1^hi^ populations [88]. CAR PD-1^hi^TIM3^hi^ TILs show a higher expression of all NR4A factors than in less exhausted CAR PD-1^hi^TIM3^lo^ populations. Besides, single-cell sequencing of CD8^+^ TILs shows a strong positive correlation between Nurr1 and PD-1 expression. Adoptive transfer of NR4A-triple knockout CAR T cells promotes tumor regression and prolongs survival in immunocompetent mice. All the three corresponding single knockout CAR T cells show inferior antitumor responses as compared to the triple knockout clones. The antitumor activity is associated with decreased PD-1 and TIM-3 gene expression and increased effector cytokine (interleukin-2 receptor α (IL-2Rα) and TNF) expression. Besides, NR4A-deficient CAR TILs have been shown to reduce chromatin accessibility to the exhaustion-inducer NFAT and increase the accessibility of binding motifs to Rel/NFκB, an initiator of T cell activation [96]. Taken together, inhibition of NR4A in TILs is a potential immunotherapeutic strategy that could possibly repress inhibitory surface receptors, increase effector cytokine production, and enrich consensus motifs associated with effector functions.

## 6. Future Perspectives

Nurr1 plays a pro-oncogenic role in different types of cancer by augmenting malignant transformation. More importantly, Nurr1 promotes self-renewal and resistance to chemotherapy and irradiation, creating a major obstacle for complete lesion clearance [16]. Targeting Nurr1 and its signaling pathway is one strategy to suppress tumor progression. Inhibition of Nurr1 by its ligands or antagonists is perhaps the most straightforward method. For instance, DIM-C-pPhCl has been shown to regress tumor formation in xenograft mouse models of glioblastoma [26] and bladder cancer [23]. Due to the limited number of Nurr1 antagonists identified, another possible approach is to block the upstream signaling that induces Nurr1 expression. Targeting COX-2 is of particular interest as it promotes the production of PGE2 and TXA2 [97]. Selective inhibition of COX-2 by parecoxib suppresses tumor growth in a preclinical study [10]. In addition, the cytoplasmic significance of Nurr1 should not be ignored as its cytoplasmic predominance correlates with several clinicopathological parameters [16,19]. Its ability to activate Akt and ERK signaling may be indicative of its oncogenic role outside the nucleus. Hence, blocking downstream signaling of Nurr1 is also an approach that is worthwhile for future exploration for abrogating Nurr1-mediated cancer aggressiveness and synergizing the chemo- or radiotherapy.

Elucidating the crosstalk of Nurr1 with oncogenic or tumor-suppressive pathways helps define key players in cancer progression. Though many interacting partners and signaling molecules downstream of Nurr1 have been reported, some of them appear contradictory. For example, many studies have reported that Akt and ERK phosphorylate Nurr1 and modulate its transcriptional activity, but recent findings identified the Akt and ERK signaling cascades as novel downstream targets of Nurr1. This provides insights into Nurr1 crosstalk and raises questions about whether there is a positive regulatory loop between Nurr1 and Akt and the mechanisms underlying Akt activation by Nurr1. 

The immunosuppressive roles of Nurr1 have been generally accepted since it was reported to be an essential mediator of Treg lineage maintenance and T cell exhaustion [80,91]. Recent findings also provide clues that Nurr1 promotes M2 macrophage repolarization [95]. Nonetheless, most of the findings are from studies conducted using different models and cannot determine the impact of Nurr1 on the dynamic immune cell profile at one time. Comprehensive and coherent immune cell profiling in cancer with and without treatment against Nurr1 is desirable. In addition, treatment with a Nurr1 antagonist or COX-2 inhibitor should be an area of active investigation as Nurr1 plays a crucial role in both cancer biology and tumor immunology. Targeting Nurr1 not only disrupts Nurr1-mediated immune tolerance, but also suppresses cancer aggressiveness. It may also be administrated synergistically with immunotherapy as a combination therapy to enhance the therapeutic effect.

## 7. Conclusions

Aberrant Nurr1 expression is associated with cancer progression and is used as a prognostic marker in several types of cancer. Multiple oncogenic pathways converge to phosphorylate NF-κB and CREB to promote Nurr1 transcription. Expression of Nurr1 is also regulated post-transcriptionally by miRNA. Nonetheless, a few studies illustrate the tumor-suppressive properties of Nurr1 and highlight a dichotomous and context-dependent role of Nurr1. Identifying the context that defines Nurr1 function and its downstream signaling provides insight into the roles of Nurr1 in promoting carcinogenesis and suppressing antitumor immunity. In conclusion, targeting Nurr1 is a potential therapeutic approach for cancer that may work by suppressing malignant phenotypes, breaking immune tolerance against cancer, and enhancing the efficacy of cancer treatments.

## Figures and Tables

**Figure 1 cancers-12-03044-f001:**
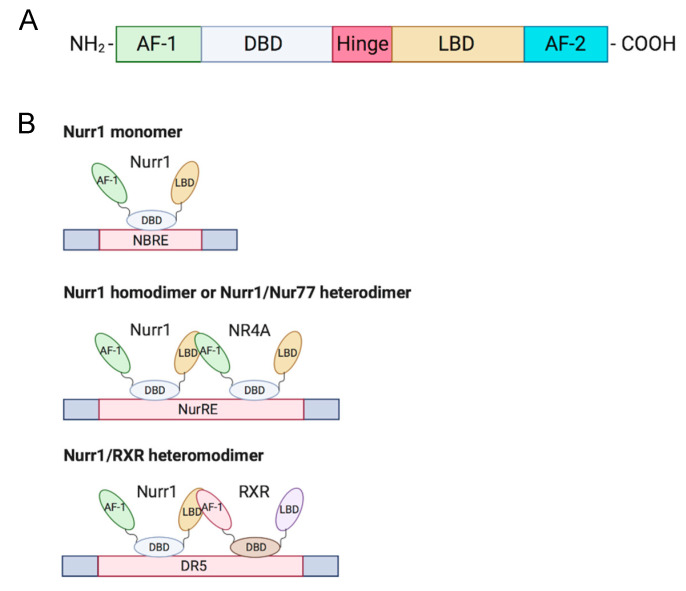
Structure and DNA-binding elements of Nurr1. (**A**) Nurr1 structure (**B**) DNA-binding elements of monomeric, homodimeric, and heterodimeric Nurr1. AF: activation function; DBD: DNA-binding domain; LBD: ligand-binding domain; NRBE: nerve growth factor-induced clone B (NGFI-B) response element; NurRE: Nur response element; DR5: direct repeat element with five spacer nucleotides element; RXR: retinoid X receptor.

**Figure 2 cancers-12-03044-f002:**
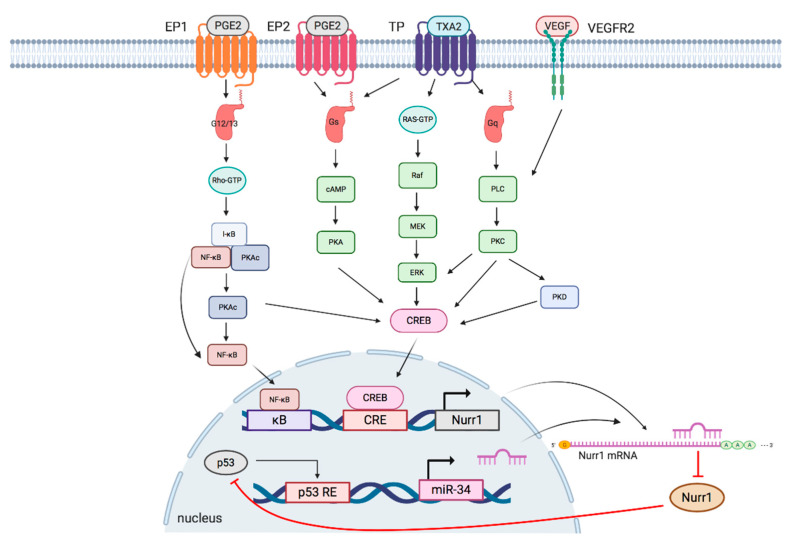
Schematic overview of signaling pathways regulating Nurr1 expression. TXA2, PGE2, and VEGF pathways modulate phosphorylation of NF-κB and CREB to regulate the transcriptional activity of Nurr1. p53/microRNA-34/Nurr1 forms a positive feedback loop to suppress p53 activation and the subsequent expression of microRNA-34, leading to the upregulation of Nurr1 protein expression. EP: prostaglandin E_2_ receptor; PGE2: prostaglandin E_2_; TP: thromboxane A_2_ receptors: TP; TXA2: thromboxane A2: VEGFR: vascular endothelial growth factor receptor; VEGF: vascular endothelial growth factor; G12/13: G_12_/G_13_ alpha subunit; Gs: G_s_ alpha subunit; Gq: G_q_ alpha subunit; κB: kappa-B; CRE: cAMP response elements; p53 RE: cAMP p53 response element.

**Figure 3 cancers-12-03044-f003:**
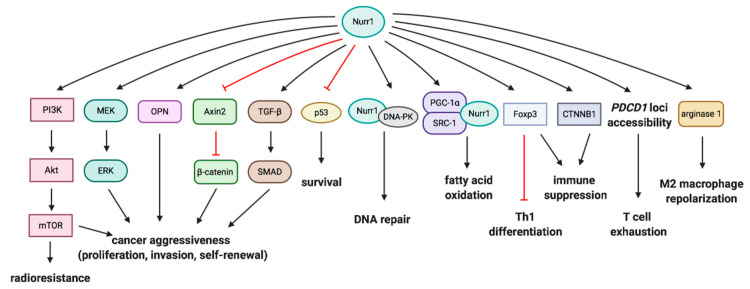
Downstream signaling of Nurr1 and the associated functions in carcinogenesis and suppression of antitumor immunity.

**Table 1 cancers-12-03044-t001:** Summary of Nurr1 expression and its associated functions and clinicopathological correlation in cancers.

Cancer	Nurr1 Expression	Functions	Stage Correlation	Survival Correlation	Ref.
Breast	Low	Inhibited tumor growth and metastasis in vivo *	No ^	Yes ^	[22]
Bladder	High	Promoted migration and tumor growth in vivo	Yes ^^^	Yes ^^^	[19,23]
Colon	High	Promoted cell proliferation, migration, and chemoresistance to 5-fluorouracil	No ^^	Yes ^^	[10,15]
Gastric	Low	Promoted apoptosis and inhibited gastrin-induced migration and invasion	*N*A	*N*A	[20,24]
	High	Promoted tumor growth in vivo and chemoresistance to 5-fluorouracil	No ^^	Yes ^^	[14]
Cervical	High	Promoted anchorage-independent growth, anoikis	*N*A	*N*A	[11,25]
Prostate	High	Promoted cell proliferation, migration, invasion, and resistance to apoptosis	Yes ^	*N*A	[12]
Pancreatic	High	Promoted cell proliferation and resistance to apoptosis	Yes ^	Yes ^	[13]
Brain	High	Promoted cell proliferation, migration, invasion and survival	NA	Yes ^#^	[26]

* Dichotomous role of Nurr1 was reported: High expression of Nurr1 in normal breast cancer in immunohistochemistry, yet silencing of Nurr1 inhibited tumor growth and metastasis in nude mice; ^ Analyzed with total Nurr1; ^^ Analyzed with nuclear Nurr1; ^^^ Analyzed with cytoplasmic Nurr1; ^#^ Analyzed with Nurr1 mRNA; NA: data not available; Ref.: reference.

**Table 2 cancers-12-03044-t002:** Signaling pathways regulating Nurr1 expression in cancer development.

Pathway	Cancer	Mode of Regulation
TXA2 pathway	Lung cancer [18]	Transcriptional regulation
PGE2 pathway	Neuroblastoma [39]; Colorectal cancer [17]; Lung cancer [41]	Transcriptional regulation
p53/miR-34/Nurr1 loop	Colorectal cancer [40]	Post-transcriptional regulation
VEGF/PKD pathway	Endothelia angiogenesis [33]	Transcriptional regulation

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
