# Peer review of "Role of Nurr1 in Carcinogenesis and Tumor Immunology: A State of the Art Review"

_cancers, 2020, doi:10.3390/cancers12103044_

Round 1

Reviewer 1 Report

Wan et al summarized and discussed the current state of the art of the Nuclear receptor related-1 (Nurr-1) protein in different (solid) cancer types, pointing out relevant molecular mechanisms related to Nurr1 expression and the subsequent readouts on cancers.  Also, the authors pointed out on the association between Nurr1 and anti-tumour (I would say T cell-restricted) immunity, discussing the possible relevance of Nurr1 as a target for immunotherapy.

The review is well written, with good quality figures summarizing the molecular interactions between Nurr1 and relevant molecular pathways in cancer biology.

Therefore, I have some comments to be addressed that would impact on the quality of the work.

1) Authors discussed the state of the art and updates on Nurr1 expression and function in solid cancer (breast, bladder, gastrointestinal, lung, cervical, prostate, pancreatic). This is a relevant effort. Therefore, even if there are very few works in the specific topic, brain cancers are missing in this complex scenario. For reference see:

10.1007/s11060-019-03349-y

10.1111/j.1471-4159.2008.05432.x

2) Section 2 “Expression and function of Nurr1 in cancers”, is totally dedicated to solid cancer. Therefore, section 5 “Nurr1 and anti-tumour immunity” discuss anti-leukemic activities. To be consistent, authors should review the state of the art of Nurr1 also in the context of haematological cancers. For references, see:

10.2174/1389450115666141120112818

10.1038/leu.2013.275

10.1182/blood.V122.21.3870.3870

10.1182/blood-2010-02-267906

10.1038/nm1579

3) In section 4, the authors discussed the “Crosstalk of Nurr1 with pro-tumorigenic and tumour-suppressive molecules. I’m very surprised that authors totally ignored Nurr1 interactions with TGFb, that acts as a master regulator both in pro-tumorigenesis, tumour suppression and immunosuppression. I think this is a relevant point to be discussed and integrated into figure 3. For references see:

10.1038/ncomms4388

The paper can be accepted, pending minor revision e an overall improvement of English quality and style.

Reviewer 2 Report

The article by Peter Kok-Ting Wan et al. is aimed to analize and decipher the role of Nuclear receptor related-1 protein (Nurr1), in carcinogenesis and tumor immunology. Nurr1 is part of the immediate early response genes and also known as NR4A2, belonging to the nuclear receptor (NR) subfamily 4 group A (NR4A) and it represents an orphan NR. Authors discussed in a clear and pertinent manner the crosstalk between Nurr1 and various signaling pathways modulating various cellular and physiological responses, and interaction of nurr1 with many oncogenic and tumor suppressor molecules, which contribute to its potential pro malignant in multiple cancers, as breast, bladder, gastrointestinal, lung, cervical, prostate and pancreatic cancers. They also discussed the involvement of nurr1 in antitumor immunity.

Overall, the review is well written, well structured and well discussed.

There a no, in my opinion major concerns.

However, minor concerns are presented here.

The authors mention the possible anticancer roles of nurr1, i.e. in the abstract at lines 16-17 it is said: “… which contribute to its potential pro malignant and tumor-protective behaviors.” Also, in Fig.3 legend it is said: “Downstream signaling of Nurr1 and the associated functions in carcinogenesis and antitumor immunity.”

However, for me it is not clear which actions are to be considered as antitumor action exerted by nurr1 in Fig. 3. Indeed, mainly if not all, the actions have pro-tumor effects.

For example, considering Th1 differentiation, authors should change the arrow to a Th1 inhibition symbol, to be correct and clearer in the message, and in red, as like those used for p53.

And so perhaps the title of chapter 5 should be changed from the current one, which is: “Nurr1 and antitumor immunity.” Maybe: Nurr1 and inhibition of antitumor immunity?

Reviewer 3 Report

Wan and et. al (Cancers 927488) has very diligently compiled the role of Nurr 1 in several cancers and also described its signaling role in various physiological processes. Authors have mentioned about expression of Nurr1 in cancer and its tumorigenic functions. Subsequently, authors have talked about how signaling pathways modulate Nurr1 expression including TXA2, PGE2, VEGF pathways. Similarly, interactions of Nurr1 with pro-tumorigenic and tumor-suppressive signaling pathways such has PI3K/Akt/mTOR, Wnt/B-Catenin, p53 etc. have been beautifully summarized. Overall, it is nicely written manuscript with only few minor issues, which must be addressed before publication.

  1. Authors have beautifully described role of Nurr1 in various cancers. In my point of view, it would be more beneficial to readers, if authors can provide role of Nurr1 in cancer biology (As a broad terminology) rather than describing each cancer type individually.
  2. By carefully reading this review, it seems like authors have summarized research articles. It will make the review greatly appreciated in the field, if authors could provide their prospective on those studies.
  3. There is a typo,

Line 196-197, “…..cell survival and growth of melanoma, prostate and urothelial cancer” (Delete “and” before prostate cancer)
